# Problematic Use of the Internet and Cybervictimization: An Empirical Study with Spanish Adolescents

**DOI:** 10.3390/bs15060810

**Published:** 2025-06-13

**Authors:** Verónica Marcos, Francisca Fariña, Manuel Isorna, Santiago López-Roel, Katia Rolán

**Affiliations:** 1Forensic Psychology Unit, Developmental and Educational Psychology Department, University of Santiago de Compostela, 15705 Santiago de Compostela, Spain; 2Faculty of Psychology, International University of Valencia, 46002 Valencia, Spain; 3AIPSE Department, UNESCO Chair on Transformative Education: Science, Communication and Society, University of Vigo, 36310 Vigo, Spain; isorna.catoira@uvigo.gal (M.I.); santiago.lopez.roel@uvigo.gal (S.L.-R.); katiarolan@uvigo.gal (K.R.)

**Keywords:** adolescence, addiction, online context, cyberbullying, victimisation

## Abstract

Background: Adolescence is a critical stage for the development of behaviours related to problematic Internet and social media use, as well as for the experience of cybervictimisation. The literature highlights the need to examine these types of adolescent behaviours. Method: A field study was designed to analyse the prevalence of problematic Internet and social media use, as well as cybervictimisation, sexting, and grooming. In total, 666 Spanish adolescents participated, 55.4% females and 44.6% males, with an age range between 14 and 18 years (*M* = 15.27, *SD* = 1.01). Results: The results showed that 15.8% of the participants were diagnosed as at risk for Internet and social media addiction; 27.2% of the participants were diagnosed as cyberbullying victims; 14.7% of the sample engaged in sexting behaviours, and 34.7% engaged in grooming behaviours. Additionally, significant gender differences were found in cybervictimisation behaviours, with females scoring higher. Conclusions: The findings are discussed in relation to the need to develop preventive and educational strategies that promote the safe and responsible use of the digital context.

## 1. Introduction

Internet use has become an essential component of daily life globally, especially among adolescents, who have grown up immersed in the digital era, using mobile devices, social networks and online games as part of their daily routine ([42]; [47]; [72]). Recent data warn of an exponential increase in the use of the Internet and social networks, which has not only modified the social interaction habits among adolescents but has also influenced the way in which they construct their identity and establish interpersonal relationships ([75]; [81]). While the digital context offers numerous educational, social, and entertainment opportunities ([85]), it also means significant risks, particularly with regard to addiction to digital technologies use ([39]). Thus, more and more young people are diagnosed with Internet addiction ([14]), and to social networks, due to immaturity in impulse control and the search for immediate gratification ([49]; [58]; [64]). This growing dependence on digital connectivity has generated concern in the scientific community due to its potential impact on the psychosocial well-being of adolescents ([28]; [70]). Internet addiction affects several neural networks that influence an adolescent’s behaviour and development ([14]), in addition to having deleterious effects at the physical, psychological and cognitive levels ([49]).

The recognition of the so-called “Internet addiction” as a possible clinical disorder has generated a wide debate in the scientific community, as it is not officially classified in the main diagnostic manuals, such as the Diagnostic and Statistical Manual of Mental Disorders (DSM-5) or the International Classification of Diseases (ICD-11) ([3]; [29]; [49]). Nevertheless, ICD-11 has recognized gaming disorder as a disorder within the category of “disorders due to addictive behaviours,” and DSM-5 has pointed out the need for further research on this condition. In this context, multiple terms have emerged to describe dysfunctional Internet use patterns, including Problematic Internet Use (PIU), Internet addiction, compulsive use, and pathological use ([68]). In the absence of a consensus on its clinical categorisation, some researchers have proposed the term PIU as a more descriptive and neutral alternative, which avoids pathologising common digital behaviours and allows for differentiating this phenomenon from other established behavioural addictions ([33]). Although there are no standardised diagnostic criteria for assessing Internet and social network addiction, several psychometric tools have been developed. Among them is the Social Networking and Internet Addiction Risk Scale (ERA-RSI) by [60] ([60]), which measures a continuum of use behaviours, from adaptive to problematic, characterised by compulsive and impulsive use that interferes with adolescents’ daily lives.

Regardless of the terminology used, the progressive concern about the negative effects of PIU has prompted numerous investigations on its impact on mental health, especially in adolescents ([4]; [69]). Several studies have evidenced that PIU is associated with increased levels of anxiety, depression, and alterations in emotional regulation, as well as lower life satisfaction and psychological well-being ([46]; [84]; [89]). The recent meta-analytic review by [74] ([74]) concludes that there is a negative relationship between Internet addiction and adolescent mental health. This underscores the need to continue exploring this phenomenon and to develop effective prevention and intervention strategies.

Estimates of PIU prevalence vary widely among studies, geographical contexts, and the methodology employed. Thus, in Asia, rates of over 25% have been reported ([68]), and in Spain, [51] ([51]), using a cut-point approach with measurement instruments based on the DSM-5, found a prevalence of 33%, but the percentage dropped to 2.98, when they took the ICD-11 as a framework. The prevalence, together with the growing evidence on its adverse effects, has prompted the need to further study its clinical manifestations, the associated risk factors and the most appropriate intervention strategies. Consequently, it is crucial to advance our understanding of PIU and to develop more homogeneous diagnostic criteria in order to differentiate between intensive Internet use and pathological use with a significant clinical impact.

A disturbing reality of the constant digital interaction of young people is the frequent exchange of messages containing sexually explicit material ([87]). In this regard, the literature has documented how exposure to digital context also increases the risk of cybervictimisation, including cyberbullying, sexting, and grooming ([43]; [44]). Because of these, cyberbullying is the most prevalent phenomenon among adolescents ([36]; [54]). Notwithstanding, there are discrepancies in its definition. Some studies consider it a form of digital violence separate from traditional bullying ([20]; [62]), while others interpret it as an extension of traditional bullying ([53]; [91]). For a correct diagnosis of cyberbullying, five fundamental criteria have been identified: violence, intentionality, repetition, power asymmetry and victimisation ([5]; [53]). These criteria also apply to related phenomena such as sexual cyberbullying, which encompasses sexting and grooming behaviours ([45]; [77]; [59]).

Regarding the effects that cybervictimisation causes in adolescents, the evidence found is alarming, not only because of the severity of these effects, including depression, anxiety, suicidal ideation, non-suicidal self-injury and substance abuse ([31]; [35]; [36]; [40]; [61]; [63]; [85]; [90]), but also because of the number of adolescents suffering from it. The results on the prevalence of cybervictimisation among adolescents vary significantly across countries, studies and measurement methods ([63]). In Spain, the reported prevalence ranges from 9% ([24]) to 39.9% ([71]). A study by [80] ([80]) estimates that in each classroom, there are approximately two students who suffer from cyberbullying.

Regarding the role of gender on cyberbullying and cybervictimisation, scientific results are not consistent ([2]; [63]; [76]). Thus, the Health Behaviour in School-aged Children (HBSC) study, conducted in several countries, found no gender differences in cyberbullying ([32]), in agreement with [38] ([38]), and contrary to [6] ([6]) who suggest that males participate more than females as bullies and in turn are more victimised. Other studies (e.g., [11]; [12]; [23]; [30]), in line with [6] ([6]), found that males engage in cyberbullying more than females; however, conversely, that females are more likely to be victims than males. [76] ([76]), in their meta-analytic study, found that females experienced only slightly more cybervictimisation. In support, [72] ([72]) only found significant differences in “Verbal–written cybervictimisation”, where girls obtained higher scores out of four types of victimization assessed. In terms of sexual victimisation on the Internet, different studies also show that females suffer it to a greater extent than males, as is the case with grooming ([66]).

Given the negative impact of problematic Internet use and cybervictimisation on adolescent well-being, and the inconsistent results on the role of gender, it is crucial to continue investigating their effects in order to develop prevention and digital education strategies to mitigate their consequences. In this context, the present study aims to analyse the prevalence of problematic Internet and social network use, as well as cybervictimisation, sexting and grooming in Spanish adolescents, also exploring gender differences and their possible interrelationship, in order to provide scientific evidence to contribute to the development of effective interventions.

## 2. Materials and Methods

### 2.1. Participants

A total of 666 adolescents participated, 55.4% females (*n* = 369) and 44.6% males (*n* = 297), with an age range between 14 and 18 years (*M* = 15.27, *SD* = 1.01). In terms of academic year, 33.0% were in the third year of Compulsory Secondary Education (14–15 years) and 38.6% in the fourth year of Compulsory Secondary Education (15–16 years), while 20.6% were in the first year of Baccalaureate (16–17 years) and 7.2% in the second year of Baccalaureate (17–18 years). The remaining 0.6% were in Intermediate Vocational Training. In terms of the type of secondary school, 70.3% were in a public school, 25.1% in a subsidised school and 4.7% in a private school in the autonomous community of Galicia (Spain).

### 2.2. Design and Procedure

A field study was designed, with a non-probabilistic accidental sampling method, in order to examine the risk behaviours of problematic use of social networks and the Internet and to examine the prevalence of victimisation by cyberbullying, sexting and grooming, as well as to analyse gender differences and the relationship between the variables under study. To obtain the sample, twelve school centres were contacted, of which nine have accepted. Once the schools confirmed their intention to participate in the study, written authorisation was obtained from each school, and informed consent was obtained from the parents or legal guardians (mandatory for those <16 years of age). Subsequently, adolescents were evaluated after also providing informed consent. The measures were administered during school hours, collectively and in a rotating order (standard rotation) in order to control the interaction between variables.

The participants completed the questionnaires, responding voluntarily, anonymously and individually, supervised by trained researchers. The collection, storage and treatment of the data were carried out in accordance with the Spanish Data Protection Act ([37]). This study was approved by the Bioethics Committee of the University of Santiago de Compostela (Code: USC54/2022).

### 2.3. Measurement Instruments

An ad hoc questionnaire was used to collect sociodemographic variables: gender, age, academic year and type of school.

To examine risk behaviours in problematic use of social networks and the Internet, the Adolescent Risk of Addiction to Social Networks and the Internet Scale [Escala de riesgo de adicción adolescente a las Redes Sociales e Internet] (ERA-RSI; [60]) was used. It consists of 29 items, with 4-point Likert-type responses (1 = Never or almost never; 2 = Sometimes; 3 = Quite often; 4 = Always), and presents four factors: addiction symptoms, social use, freak traits, and nomophobia. The addiction symptoms factor contains nine items that check behaviours of addiction to non-toxic substances (e.g., “I access a social networks and Internet anywhere and at some time”, “Right now I would feel angry if I had to do without social networks and Internet”, “I have lost hours of sleep by connecting to social networks and watching series”). The main social factor is made up of eight items and evaluates habitual adolescent virtual socialisation behaviours (e.g., “I check my friends’ profiles”, “I use the chat”, “We comment on photographs among friends”). The freak traits factor, with six items, includes aspects such as joining groups with specific interests, playing virtual and role-playing games and having sexual encounters. The nomophobia factor groups within six items related to anxiety and control in cell phone use (e.g., “If my messages are not answered immediately, I feel anxious and distressed”, “I would be furious if my cell phone were taken away”). The questionnaire presents adequate reliability indices, both for the total scale (α = 0.90) and by factors (α = 0.81 addiction symptoms; α = 0.82 social use; α = 0.72 freak traits; α = 0.82 nomophobia), and offers good internal consistency (α = 0.86) (>0.80; [52]).

To assess cyberbullying, the European Cyberbullying Intervention Project Questionnaire (ECIPQ; [19]) was used. This instrument presents a two-dimensional perspective of bullying (the person bullying vs. the person being bullied). It is composed of 22 items (e.g., “Someone has said bad-sounding words to me,” “I have said bad-sounding words to someone”), on a 5-point Likert-type scale for frequency (1 = Never or almost never happens to me; 2 = Once a month; 3 = Two or three times a month; 4 = Once a week; 5 = Several times a week). Additionally, for a correct diagnosis of bullying, in the case of a positive response and frequency greater than two or three times a month or more, it asked about the periodicity (diagnostic criterion of chronicity of bullying) with which the bullying had occurred, on a 4-point Likert-type scale (1 = Up to 1 month; 2 = Up to 3 months; 3 = Up to 6 months; 4 = More than 6 months). It consists of two factors: cybervictimisation and cyberaggression, both with good reliability indices, α = 0.80 and α = 0.88, respectively. The internal consistency found for the total scale was α = 0.87. For consideration of addiction risk, conversion of direct total scale scores to percentiles was calculated for adolescent females and males (scores ≥ to the 95th percentile indicate very high risk of addiction to Internet and social networks; scores ≥ to the 85th percentile signal addiction risk; a score ≥ to the 75th percentile opens suspicion of addiction risk). In the present study, the cybervictimisation factor is examined, which presented a good reliability index (α = 0.80) (>0.80; [52]).

Sexting and Grooming Scales were used ([59]). The Sexting scale is composed of 13 items on a 4-point Likert-type scale, ranging from 1 (Never) to 4 (Always), with a bifactor structure. The first factor, composed of nine items, has been termed erotic sexting (α = 0.87). The second factor, composed of four items, has been denoted pornographic sexting (α = 0.79). Behind each of these questions comes a list of adjectives or phrases that indicate a certain eroticisation and sexualisation of virtual postings. The full sexting scale presents an internal consistency of 0.86. The Grooming Scale is composed of 13 items with a unifactorial structure (e.g., Do you know someone who shows his or her naked body?). The total scale presents good reliability indices (α = 0.86). For the interpretation of the diagnosis of sexting, the conversion of direct scores of the total scale into percentiles was calculated, taking into account the scales according to gender and age (>95 high risk of erotic and pornographic sexting; 85–94 risk of erotic and pornographic sexting; 75–84 alert to the risk of erotic and pornographic sexting; <74 are not considered risk scores). For the interpretation of the grooming diagnosis, the conversion of direct scores of the total scale into percentiles was calculated in adolescent females and males (>95 high risk of grooming; 85–94 risk of grooming; 75–84 alert to risk of grooming; <74 not considered risk scores). In this sample, consistency levels of 0.85 and 0.84 were obtained for Sexting and Grooming, respectively (>0.80; [52]).

### 2.4. Data Analysis

Descriptive analyses and frequency analyses were calculated for the variables under study. To estimate gender differences, Student’s *t*-tests were performed for independent samples. To determine the effect size, Cohen’s *d* was analysed, whose parameters were determined between 0.20 (small), 0.50 (medium) and 0.80 (large) ([15]). To try to deepen the analysis of the relationship between addiction risk behaviours and cybervictimisation experiences, Pearson correlations (*r*) were performed. Additionally, the reliability (internal consistency) of the measurement instruments in the present study sample was calculated using Cronbach’s α coefficient. All analyses were performed using the IBM SPSS Statistics version 29 statistical software.

## 3. Results

### 3.1. Internet and Social Networks Addiction Behaviours

The results showed that 91% of the sample started using social networks before the age of 14, with Instagram (42.5%) and WhatsApp (35.1%) being the first social networks used. Likewise, 71.8% reported having consulted an adult when they first started using social networks, compared to 28.2% who indicated that they had not previously consulted an adult. Prevalence of the different addiction risk behaviours of adolescents to social networks and the Internet (addiction symptoms, social use, freak traits and nomophobia) was examined. The results revealed that 15.8% (*n* = 105), 95% CI [0.248, 0.366], of the participants were diagnosed with risk of “Internet and Social Networking addiction” (reliability, α = 0.80). The results of the descriptive analyses indicated mostly mean values for addiction risk behaviours, with a response range from 1 (Never) to 4 (Always) (see Table 1). In terms of gender, no significant differences were obtained between females and males regarding the addiction risk to social networks and the Internet (*p* > 0.05).

### 3.2. Online Victimisation: Cyberbullying, Sexting and Grooming

Second, 27.2% (*n* = 165), 95% CI [0.237, 0.305], of the participants were diagnosed as victims of cyberbullying (1 or more actions involving intentionality; repeated over time—≥1 time per week; and chronic—the bullying actions lasted 3 or more months). Regarding gender differences, significant differences were found in cyberbullying victimisation behaviours, *t*(653) = 2.25, *p* < 0.05, with females obtaining higher scores, with a small effect size (*d* = 0.18).

The results corroborated that 14.7% (*n* = 98) of the participants presented sexting behaviours (reliability, α = 0.85). Regarding gender differences, significant differences were obtained in sexting behaviours, *t*(664) = 2.10, *p* < 0.05, with higher scores obtained by females (*M* = 19.29, SD = 5.11) versus males (*M* = 18.39, *SD* = 5.96), with a small effect size (*d* = 0.16). On the other hand, sexting behaviours were examined, yielding low median values for the study population, with a response range from 1 (Never) to 4 (Daily) (see Table 2).

The results indicated that 34.7% (*n* = 231) of the participants revealed grooming behaviours (reliability, α = 0.85). Regarding gender differences, significant differences were found in grooming behaviours, *t*(664) = 3.13, *p* < 0.05, with females (*M* = 17.98, *SD* = 5.29) scoring higher than males (*M* = 16.79, *SD* = 4.56), with a small effect size (*d* = 0.24). Grooming behaviours were analysed, finding low-average values for the study population, with a response range from 1 (Never) to 4 (Daily) (see Table 3).

With regard to the correlations between the variables under study (see Table 4), significant and direct relationships were found between addiction risk factors and cybervictimisation experiences (i.e., cyberbullying, sexting and grooming).

## 4. Discussion

Adolescence is a critical period in the development of risk behaviours, including PIU and social networking, as well as cybervictimisation in the digital context ([49]). In the international context, the results of research conducted in recent years warn that Internet addiction is one of the most important mental health problems, and attention should be focused on it ([50]). In this context, the present study examined the prevalence and factors associated with the risk of Internet and social network addiction and cybervictimisation behaviours in adolescents. Notwithstanding, the findings should be interpreted taking into account certain methodological limitations.

First, the sample used was not probabilistic, which makes it difficult to extrapolate the findings to the general population. Although this type of sampling is common in studies with adolescents, future research could consider more representative designs to improve the generalizability of the results. Second, causal relationships between the variables evaluated cannot be established from the study. This is especially relevant, given that previous research has suggested that some relationships between risk behaviours may be bidirectional ([17]), albeit the mechanisms underlying these associations still require further analysis. In the case of PIU and cybervictimisation, a longitudinal design would allow us to examine the evolution of these phenomena and their possible interactions over time. Third, self-report instruments have been used, which is frequent in this type of research, which can generate social desirability or denial biases in the responses ([22]); although it has also been evidenced that they can be reliable and even more effective than other techniques ([8]; [88]). Moreover, the guaranteed anonymity and voluntary participation in this study help to reduce these biases. Fourth, the assessment of Internet addiction risk and cybervictimisation was based solely on psychometric measures, without the support of clinical interviews or other assessment records. This may affect the accuracy in detecting problem behaviours and underscore the importance of employing complementary methods in future research. Finally, other key mental health variables, such as emotional well-being, stress regulation or resilience, which could influence the relationship between PIU and cybervictimisation, were not included. Despite these limitations, this study contributes significantly to knowledge about the prevalence and factors associated with the Internet risk and social network addiction and cybervictimisation behaviours in adolescents.

First, results revealed that young people used the Internet and social networks before the age of 14 (91%), with Instagram being the preferred platform (42.5%), in line with previous studies ([78]; [86]). Likewise, it was found that 28.2% of the participants had not asked permission from an adult to use the platforms. This finding confirms that the use of the Internet and social networks is common among adolescents, and consequently, adolescents should be monitored and informed about the risks that occur in the online context from the family and educational context ([73]). Secondly, it was found that 15.8%, 95% CI [0.248, 0.366] of adolescents presented a risk of “Internet and Social Network addiction”, with no significant differences between females and males. This percentage is lower than the 25% found by [68] ([68]) with the Asian population and the 20.4% obtained by [50] ([50]) with Indian dental students, but it is nonetheless highly worrying. Thus, there is a need to pay close attention to risk and protective factors associated with the risk of addiction ([1]; [21]), both in females and males, given that adolescence is a stage where self-regulation skills are still developing, and these are relevant in the use and abuse of the Internet and social networks ([56]). Third, the study findings indicate that 27.2%, 95% CI [0.237, 0.305] of the participants suffer cyberbullying [1 or more actions involving intentionality; repeated over time (>1 time per week)]; and chronic [bullying actions lasted 3 or more months], finding significant differences in victimisation behaviours as a gender function. This percentage, although lower than the 39.9% established by [71] ([71]), is tremendously alarming, considering the negative consequences for young people ([31]; [35]; [36]; [40]; [63]; [85]; [90]). Furthermore, if it takes into consideration that there is evidence on the propensity of adolescents to not disclose cybervictimisation experiences ([10]), it could be thought that the percentage of adolescents who suffer some type of cybervictimisation could be even higher.

In terms of gender and cybervictimisation, the results show significant gender differences, with females having higher levels of cybervictimisation, albeit with a small effect size. This is in line with the meta-analytic study by [76] ([76]), in that females experienced only slightly more cybervictimisation than males, which is also consistent with previous studies that have noted that adolescent females may be more vulnerable to certain forms of digital victimisation, such as non-consensual sexting and bullying on social networks, due to factors such as gender dynamics and greater exposure in the digital context ([34]). Likewise, [9] ([9]) suggest that this difference can be explained because females tend to engage more frequently in relational forms of aggression than in physical forms of aggression, which can be transferred to the digital context and amplified through social networks. In addition, patterns of use of online platforms differ between genders, which could increase their exposure and vulnerability to cyberbullying ([7]). Nevertheless, the mechanisms underlying these differences are not yet fully elucidated, so future research should explore in greater depth gender dynamics and psychosocial factors that influence cyberbullying in adolescents. Similarly, it was observed that 14.7% of participants reported having engaged in sexting behaviours, and 34.7% in grooming behaviours, with girls scoring higher on both phenomena, albeit with a small effect size in both cases. These are more than disturbing results, in line with other studies ([57]; [66]), which justifies concern at both scientific and societal levels ([26]).

Thus, with a prevention view, adolescents should be informed about risky online behaviours, the negative effects of cybervictimisation and the support they can receive if they are involved in such experiences, since prevention programs implemented at school are effective in preventing and reducing cybervictimisation experiences ([13]; [27]; [79]). In addition, in order to promote prevention, teaching coping strategies such as seeking social support, assertive communication, and emotional management is essential, as well as informing about legal mechanisms for reporting and protecting privacy ([65]; [48]; [67]). Empowering victims and fostering their resilience allows them to take control of the situation and reduce the long-term psychological impact, avoiding revictimisation ([55]; [18]). Likewise, the need to train families about the risks associated with these problems is also highlighted ([34]; [82]). Since family dysfunction affects cyberbullying and cybervictimisation ([90]), families should also be made aware of their responsibility to protect, educate, accompany and support their children in a positive way and specifically in the responsible and safe use of the Internet and social networks. Furthermore, to a greater extent, when children suffer any type of cybervictimisation, including cases derived from previous sexting or grooming behaviour, it is in these circumstances that they are more vulnerable to engage in risky behaviours.

In conclusion, the results of this study reinforce the idea that adolescence is a critical period for vulnerability to Internet and social network addiction. Therefore, is imperative, in line with [25] ([25]), [41] ([41]) and [83] ([83]) to create, develop, and implement action plans that are based on (1) informing about the risks and consequences of uncontrolled use on the social media; (2) paying attention to risk and protective factors in this matter; (3) encouraging from the educational and family context healthy habits during this stage of development of the individual ([16]).

Taking into account the growing impact of PIU on adolescent well-being, future studies should address the relationship between PIU and cybervictimisation from a longitudinal approach, integrating psychosocial variables to better understand the evolution of these behaviours and establish more effective strategies for prevention and intervention.

## 5. Conclusions

Adolescence is a critical stage in a person’s development, as it involves numerous physiological and psychosocial changes. These changes in social, emotional, and cognitive processes, in turn, affect the ability to develop communication skills, establish relationships with others, and regulate one’s emotions. They also influence thoughts, attitudes, and beliefs, and similarly impact decision-making and impulse control. At the same time, the family and school play an important role in a person’s developmental competence during adolescence, as they contribute to the formation of socio-emotional and communication skills, as well as the acquisition of adult roles and responsibilities.

Due to excessive use of new Information and Communication Technologies during childhood and adolescence, changes have occurred in the way young people relate and communicate, influencing their ways of thinking and affecting mental health from an early age. This reinforces the need to promote scientific evidence on problematic Internet and social media use, as well as the potential for addiction to these technologies. Specifically, this study highlights the risk of Internet addiction that can emerge during adolescence. Additionally, it raises concerns about the emergence of phenomena linked to specific practices among adolescents, such as sexting, grooming, and online abuse.

In short, adolescence is a key stage at which to focus preventive action plans on digital safety within educational, family, and community settings.

## Figures and Tables

**Table 1 behavsci-15-00810-t001:** Descriptive analyses of addiction risk behaviours.

	Mean (*M*)	Standard Deviation (*SD*)
Considering all the times I visit social networks and WhatsApp, not related to studying, the time I spend on them daily is: (1) Around 1 h (2) About 2 h (3) Between 3 and 4 h (4) More than 4 h.	2.66	1.02
I use social networks during my study hours.	2.13	0.75
Right now, I would feel angry if I had to give up social networks.	2.29	0.97
I access social networks anywhere and at any time.	2.62	0.97
I believe that using social networks has interfered with my academic work.	2.03	0.97
I have lost hours of sleep due to using social networks and watching series.	2.36	1.04
I hide the time I spend on social networks at home.	1.57	0.81
If I don’t have access to the internet, I feel insecure.	1.61	0.79
I update my status.	1.79	0.84
I check my friends’ profiles.	2.53	0.89
I use the chat.	3.32	0.82
I upload photos and/or videos.	2.34	0.99
I comment on photos with friends.	2.61	1.02
I see what my contacts have been doing in the past few hours.	1.77	0.91
I find old friends.	2.34	0.89
The number of photos I have posted on social networks and the Internet is: (1) <100; (2) 101 to 1000; (3) 1001 to 3000; (4) >3000	1.17	0.45
I make new friends.	2.29	0.93
I play virtual and/or role-playing games.	2.42	1.07
I join interest groups.	1.77	0.89
I search for information about sexuality.	1.80	0.91
I visit erotic websites.	1.65	0.86
I have sexual meetings.	1.13	0.43
I use my phone to write erotic messages.	1.27	0.55
I feel safer or more accompanied knowing that I can communicate with someone at any time.	2.71	0.93
I believe it’s safer to send a photo via phone than to post it on other social networks.	2.28	0.98
If they don’t respond to my messages immediately, I feel anxiety and distress.	1.60	0.76
If they don’t respond to my messages immediately, I feel anxiety and distress.	2.04	0.91
I would get angry if they took away my phone.	2.05	0.91
I get anxious if no one talks to me when I am online.	1.61	0.83

**Table 2 behavsci-15-00810-t002:** Descriptive analyses of sexting behaviours.

		Mean (*M*)	Standard Deviation (*SD*)
I choose photos to post on social networks in which I look:	Attractive	2.66	1.02
	Sexy	2.13	0.75
	Romantic	2.29	0.97
	Erotic	2.62	0.97
	Interesting	2.03	0.97
	Provocative	2.36	1.04
	Seductive	1.57	0.81
	Sensual	1.61	0.79
	Suggestive	1.79	0.84
Among friends, we send photos in which I am:	Almost pornographic	2.53	0.89
	In underwear	3.32	0.82
	Without clothes	2.34	0.99
	Upper or lower torso nude	1.61	0.83

**Table 3 behavsci-15-00810-t003:** Descriptive analyses of grooming behaviours.

Grooming Behaviours	Mean (*M*)	Standard Deviation (*SD*)
Do you know anyone who shows their naked body on social networks and the internet?	2.07	1.06
Do you know people your age who masturbate in front of the webcam?	1.28	0.63
Have you shown your genitals or naked body on the internet?	1.07	0.33
Have you ever masturbated in front of the webcam while you or the other person was in underwear?	1.07	0.35
Have you had any conversation via webcam while you or the other person were in underwear?	1.14	0.47
Do you use the webcam to seduce or have erotic conversations with someone else?	1.06	0.31
I have sent sexy and erotic photos with the intention of flirting.	1.19	0.51
I have sent text messages with suggestive and erotic content.	1.44	0.70
I have received somewhat daring and sexual comments online.	1.66	0.82
Have you ever been asked to show yourself naked or to show your genitals on Social Networks and the Internet?	1.48	0.81
I have been sent erotic photos to seduce me.	1.63	0.85
Has any adult tried to meet with you with unclear affectionate intentions?	1.20	0.57
Has someone you didn’t know tried to convince you, give you a gift, or offer you money in exchange for showing your body online?	1.15	0.48

**Table 4 behavsci-15-00810-t004:** Correlations between the variables under study.

	1.	2.	3.	4.	5.	6.
1. Addiction symptoms	–					
2. Social use	0.574 **	–				
3. Freak traits	0.294 **	0.226 **	–			
4. Nomophobia	0.522 **	0.441 **	0.218 **	–		
5. Cyberbullying	0.284 **	0.264 **	0.207 **	0.247 **	–	
6. Sexting	0.393 **	0.455 **	0.353 **	0.300 **	0.256 **	–
7. Grooming	0.349 **	0.359 **	0.359 **	0.234 **	0.434 **	0.483 **

Note. ** Correlation is significant at 0.01 level (bilateral).

## Data Availability

The datasets used and/or analysed during the current study are available from the corresponding author on reasonable request.

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
