# Peer review of "Problematic Use of the Internet and Cybervictimization: An Empirical Study with Spanish Adolescents"

_behavsci, 2025, doi:10.3390/bs15060810_

Round 1
Reviewer 1 Report
Comments and Suggestions for Authors
The manuscript is well-structured and well-written. It provides valuable information about gender differences, types of cyberbullying, and Internet addiction during adolescence. The study's findings can be used to design prevention and educational programs for Internet addiction and reduce cyberbullying.
However, the munscript needs the following improvements:
Introduction
- A better explanation of adolescence is a critical developmental stage for developing Internet addiction and other forms of cyberbullying.
- Gender differences across different types of online victimization should be added using more empirical evidence.
- The authors do not clearly mention their hypothesis.
Methods
This part needs significant improvement.
4. The authors need to add more details about the selection of the schools. What selection criteria were used to approach the schools? Is it a randomized selection or not? If it is not, explain the reason for other types of sample/school selection.
5. The rate of drop-outs should be added.
6. More demographic information about the school sample will be useful.
7. Data collection should be explained in more detail (e.g., the duration of the study, the onset period of the study, the duration of the administration, how and where students complete the questionnaires).
8. The plan of analysis needs to be described more clearly. T-test analysis or correlations were not mentioned in the Methods section.
Results
The results were not presented clearly.
9. Sample demographic information needs to be presented.
10. Gender differences were not illustrated clearly in the tables. It is recommended to use bar charts so they can easily present the gender differences of online victimization types.
Discussion
11. The strengths and the limitations of the study are very general. They should be more specific for the current study.
12. Theoretical and practical implications of the study should be analyzed in more detail.
Author Response
Comments 1: Introduction. 1. A better explanation of adolescence is a critical developmental stage for developing Internet addiction and other forms of cyberbullying. 2. Gender differences across different types of online victimization should be added using more empirical evidence. 3. The authors do not clearly mention their hypothesis.
Response 1: Thank you for pointing this out. We agree with this comment. Information has been added regarding the first point. Regarding the hypotheses, it is specified in the last paragraph of the introduction that the main objective is to analyze the prevalence.
Comments 2: Methods. This part needs significant improvement. 4. The authors need to add more details about the selection of the schools. What selection criteria were used to approach the schools? Is it a randomized selection or not? If it is not, explain the reason for other types of sample/school selection. 5. The rate of drop-outs should be added. 6. More demographic information about the school sample will be useful. 7. Data collection should be explained in more detail (e.g., the duration of the study, the onset period of the study, the duration of the administration, how and where students complete the questionnaires). 8. The plan of analysis needs to be described more clearly. T-test analysis or correlations were not mentioned in the Methods section.
Response 2: Thank you for pointing this out. Thank you for this clarification. However, this information is included in each section of the method.
Comments 3: Results. The results were not presented clearly. 9. Sample demographic information needs to be presented. 10. Gender differences were not illustrated clearly in the tables. It is recommended to use bar charts so they can easily present the gender differences of online victimization types.
Response 3: Thank you for pointing this out. Thank you for this suggestion. However, it was decided not to include graphs to avoid redundancy in the information.
Comments 4: Discussion. 11. The strengths and the limitations of the study are very general. They should be more specific for the current study. 12. Theoretical and practical implications of the study should be analyzed in more detail.
Response 4: Thank you for pointing this out. We agree with this comment. A final section of conclusions has been added.
Reviewer 2 Report
Comments and Suggestions for Authors
Dear authors, I made some suggestions and hope they will help improve your work.
Concept definitions review:
In the abstract and introduction, the description of the grooming concept might lead to misunderstanding or misinterpretation. For instance, the statement indicating that 34.7% of adolescents engaged in grooming-related behaviours requires clarification. Specifically, it should be made clear whether these behaviours involved adolescents interacting among themselves, or if they involved adults manipulating minors for sexual purposes. It is necessary to clarify whether adolescents were potential victims (exposed to grooming) or if they were reporting awareness of grooming behaviour among peers. I recommend revising these statements in both the abstract and introduction.
Additionally, the abstract should briefly mention practical implications of the results, such as specific prevention or intervention strategies outlined in the discussion
Methodology:
In the "Measurement Instruments" section, the description of instruments is detailed and well-presented. Nevertheless, I recommend including a concise table or figure summarising key details of the scales employed (for example, indicators studied, type of scale, authorship, and a brief description). This would improve clarity and facilitate quick reference to the methods used.
Results:
In Table 1, please revise the phrase “Si no responden inmediatamente a mis mensajes siento ansiedad y angustia”. It should be presented only in English to maintain the previously recommended linguistic consistency.
Discussion:
I recommend presenting the study limitations (lines 287 to 309) separately, following the main discussion and before the conclusions. It is unnecessary to introduce new headings; reorganise the content to ensure that the limitations are not the first point addressed in the discussion, thereby improving the logical flow of the manuscript.
Comments on the Quality of English LanguageLinguistic consistency:
It is essential to revise the document to ensure linguistic consistency throughout. Different spelling variations have been identified, such as "behavior" and "behaviour". I suggest selecting one form consistently throughout the manuscript, preferably British English ("behaviour"), to ensure uniformity.
Author Response
Comments 1. Methodology: In the "Measurement Instruments" section, the description of instruments is detailed and well-presented. Nevertheless, I recommend including a concise table or figure summarising key details of the scales employed (for example, indicators studied, type of scale, authorship, and a brief description). This would improve clarity and facilitate quick reference to the methods used.
Response 1: Thank you for pointing this out. Thank you for this suggestion. However, it was decided not to include table to avoid redundancy in the information.
Comments 2. Results: In Table 1, please revise the phrase “Si no responden inmediatamente a mis mensajes siento ansiedad y angustia”. It should be presented only in English to maintain the previously recommended linguistic consistency.
Response 2: Thank you for pointing this out. We agree with this comment. The modification has been made.
Comments 3. Discussion: I recommend presenting the study limitations (lines 287 to 309) separately, following the main discussion and before the conclusions. It is unnecessary to introduce new headings; reorganise the content to ensure that the limitations are not the first point addressed in the discussion, thereby improving the logical flow of the manuscript.
Response 3: Thank you for pointing this out. Thank you for this suggestion. We chose to present the limitations first so that when the results are discussed, they are kept in mind in terms of generalization and extrapolation.
Reviewer 3 Report
Comments and Suggestions for Authors
I sincerely enjoyed reading the present paper “Problematic use of the Internet and cybervictimization: an empirical study with Spanish adolescents,” which analyzes “the prevalence of problematic Internet and social network use, as well as cyber-victimization, sexting, and grooming in Spanish adolescents” and also “exploring gender differences and their possible interrelationship, in order to provide scientific evidence to contribute to the effective interventions development.” I wish to congratulate my colleagues for their painstaking effort in providing us with a scientifically grounded and timely paper that contributes to the study of Problematic Internet Use (PIU) and cybervictimization. The authors managed, despite methodological limitations, to offer valuable findings and formulate their recommendations to stakeholders for prevention. While I find the manuscript well-organized, with a clear abstract, introduction, methods, results, and discussion, I can observe that perhaps the excessive detail in terminology slightly reduces the ease of reading. I am also particularly pleased with the rich and, above all, recent bibliography (2017-2025), the reliable psychometric tools (e.g., ERA-RSI, ECIPQ, etc.) with high reliability that enhance the study’s validity, and the appropriate statistical analyses used (t-tests, Pearson correlations, Cohen’s d). However, the paper is limited by non-probabilistic accidental sampling and small effect sizes, which reduce its clinical significance {e.g., (Cohen’s d) gender differences, although present, are not large enough to be considered critical on their own}, indicating the need for further research to explore other factors that may more strongly influence these behaviors. The authors, of course, could not but:
- propose educational/preventive strategies, which, in my opinion, need greater specificity for immediate application,
- acknowledge limitations (e.g., self-reports, lack of longitudinal analysis), and
- suggest future directions.
However, they do not discuss potential cultural peculiarities of Spain that may influence the results, which could have enriched their analysis.
Considering the above concise report, I wish to propose to my colleagues the following optional improvement suggestions:
- Streamline the introduction, focusing on the term “PIU” and avoiding excessive detail on terminology.
- Strengthen intervention proposals with specific examples or program models.
- Discuss cultural or social factors in Spain that may influence the findings.
- Consider the use of longitudinal designs or clinical interviews in future studies for stronger conclusions.
Author Response
Comments 1. Considering the above concise report, I wish to propose to my colleagues the following optional improvement suggestions: 1. Streamline the introduction, focusing on the term “PIU” and avoiding excessive detail on terminology. 2. Strengthen intervention proposals with specific examples or program models. 3. Discuss cultural or social factors in Spain that may influence the findings. 4. Consider the use of longitudinal designs or clinical interviews in future studies for stronger conclusions.
Response 1: Thank you for pointing this out. Thank you for this suggestion. I would like to thank you for your review, as well as for all the suggestions provided to improve the manuscript. They will be taken into account when making changes. Regarding the suggested recommendations, it will be taken into account to provide more detailed intervention proposals, as well as to discuss cultural or social factors. Regarding the introduction, the term PIU is detailed along with its specific definition to provide a better explanation of the phenomenon. Likewise, the suggestion to strengthen the intervention section has been taken into account, and as a result, a conclusions section has been added to further address this issue. Concerning the final suggestion, indeed, our study is cross-sectional in nature, which may be a limitation in the interpretation of the results. However, longitudinal designs are recommended for future studies, and this will be considered in upcoming research. We agree that the design of longitudinal studies should be considered. All the suggested recommendations are appreciated.
Reviewer 4 Report
Comments and Suggestions for Authors
Thank you for the opportunity to review this interesting manuscript that examines problematic Internet use and digital victimization among adolescents. The study addresses important public health issues and includes a large and relevant sample. Below are some recommendations for improving the manuscript:
- The introduction provides a comprehensive review of the literature on PIU and cybervictimization. However, although comprehensive, it could benefit from a more focused articulation of the study's rationale. In particular, it should be clarified how this study advances knowledge beyond the prevalence statistics already available in Spanish or European datasets. Also, key constructs (e.g., "freak traits") should be defined earlier and more precisely. The current explanation appears later in the methods section and is unclear to an international audience.
- The use of non-probabilistic accidental sampling is a significant limitation of the study. Please briefly explain why this method was chosen and discuss its implications for generalizability.
- Pearson correlations, t-tests, and descriptive statistics were used. However, regression models would have improved the analysis by testing associations between variables while controlling for confounding factors (e.g., age, gender, and school type).
- Given the prevalence of behaviors such as grooming (34.7%) and sexting (14.7%), discuss whether these levels are consistent with existing data in Spain or the EU, and if not, why this might be the case.
- Include a more detailed discussion of potential mediators and moderators (e.g., family dynamics, peer influence) that may explain the co-occurrence of PIU and cybervictimization, with reference to longitudinal or network-based models.
- Explicitly address the directionality issue raised in your limitations. Consider proposing future longitudinal or cross-lagged panel studies to clarify temporal precedence.
- The recommendation to strengthen families and schools is important, but general. Propose specific intervention models.
- The literature review could be expanded to include recent findings on the role of parental mediation and family dynamics in adolescents' digital behaviors. Specifically, I recommend that the authors cite and discuss these recent articles on the topic to strengthen this section of the manuscript: 10.1111/jora.13034; 10.1016/j.psychres.2017.05.030; 10.1007/s12144-023-04557-6.
A careful proofreading of the manuscript is recommended to improve readability.
Author Response
Comments 1. Below are some recommendations for improving the manuscript: 1. The introduction provides a comprehensive review of the literature on PIU and cybervictimization. However, although comprehensive, it could benefit from a more focused articulation of the study's rationale. In particular, it should be clarified how this study advances knowledge beyond the prevalence statistics already available in Spanish or European datasets. Also, key constructs (e.g., "freak traits") should be defined earlier and more precisely. The current explanation appears later in the methods section and is unclear to an international audience. 2. The use of non-probabilistic accidental sampling is a significant limitation of the study. Please briefly explain why this method was chosen and discuss its implications for generalizability. 3. Pearson correlations, t-tests, and descriptive statistics were used. However, regression models would have improved the analysis by testing associations between variables while controlling for confounding factors (e.g., age, gender, and school type). 4. Given the prevalence of behaviors such as grooming (34.7%) and sexting (14.7%), discuss whether these levels are consistent with existing data in Spain or the EU, and if not, why this might be the case. 5. Include a more detailed discussion of potential mediators and moderators (e.g., family dynamics, peer influence) that may explain the co-occurrence of PIU and cybervictimization, with reference to longitudinal or network-based models. 6. Explicitly address the directionality issue raised in your limitations. Consider proposing future longitudinal or cross-lagged panel studies to clarify temporal precedence. 7. The recommendation to strengthen families and schools is important, but general. Propose specific intervention models. 8. The literature review could be expanded to include recent findings on the role of parental mediation and family dynamics in adolescents' digital behaviors. Specifically, I recommend that the authors cite and discuss these recent articles on the topic to strengthen this section of the manuscript: 10.1111/jora.13034; 10.1016/j.psychres.2017.05.030; 10.1007/s12144-023-04557-6.
Response 1. I would like to thank you for your review, as well as for all the suggestions provided to improve the manuscript. They will be taken into account when making changes. All suggestions for improvement are appreciated. Regarding the suggested recommendations:
- We appreciate the acknowledgment of the comprehensive review of the literature on problematic Internet use (PIU) and cybervictimization. The specific contribution of this manuscript is discussed in the discussion section. As for key concepts, they are detailed in the section on measurement instruments to avoid confusion with terms used by other researchers.
- The study design may be one of its limitations and, therefore, it is mentioned in the discussion section. Additionally, the "design and procedure" section describes the research process and the aspects that were taken into account.
- We appreciate the suggested types of analyses, and they will be considered for future manuscripts. In this case, the focus was placed on prevalence data, which is why these specific analyses were selected.
- Potential moderating variables will be considered for future research and, for this reason, this issue has also been added to the discussion section. Moreover, a "conclusions" section has been added to the manuscript for greater depth.
- We are grateful for the suggested references, and they have been incorporated into the manuscript.
Once again, thank you for all the suggestions to improve the manuscript.
Round 2
Reviewer 4 Report
Comments and Suggestions for Authors
The paper is significantly improved and I have no further revisions to suggest.